# Thermal Kinetics of a Lignin-Based Flame Retardant

**DOI:** 10.3390/polym12092123

**Published:** 2020-09-17

**Authors:** Xiaoxuan Liang, Qixiang Hu, Xu Wang, Liang Li, Yuguo Dong, Chang Sun, Chengjuan Hu, Xiaoli Gu

**Affiliations:** Co-Innovation Center for Efficient Processing and Utilization of Forest Products, College of Chemical Engineering, Nanjing Forestry University, Nanjing 210037, China; xiaoxuanliang1998@outlook.com (X.L.); hu_qixiang@outlook.com (Q.H.); xuWang2020@outlook.com (X.W.); liliang250035@gmail.com (L.L.); dongyuguo99@outlook.com (Y.D.); 18851942018sc@sina.com (C.S.); hcj180203406@hotmail.com (C.H.)

**Keywords:** lignin, flame retardant, kinetic analysis

## Abstract

In order to improve the thermal property of epoxy resin (EP), a lignin-based flame retardant was prepared. Focusing on the lignin-based flame retardant, this paper investigates its pyrolysis behavior and kinetics via a thermogravimetric analyzer coupled with Fourier transform infrared spectrometry (TG–FTIR). Based on the FTIR result, which showed a peak at 1222 cm^−1^, it was assigned a syringyl structure. Its absorption peak intensity was enhanced and this meant that the phenolization of the lignin was successful. Thermogravimetry/derivative thermogravimetry (TG/DTG) results showed that the carbon residues of F-lignin and F-lignin@APP were reduced to 33.5% and 37.5%, respectively. In addition, the maximum decomposition rate of F-lignin@APP20/EP is 11.8%/min, which is 8%/min and 4.7%/min lower than for EP and Al-lignin, respectively. The char residue of F-lignin@APP20/EP is 32.5%, which is much higher than for EP. Lower decomposition rate and higher char residue indicate the improvement of thermal stability of EP by F-lignin@APP. Moreover, the kinetics of Al-lignin20/EP and F-lignin@APP20/EP were conducted by two kinetic methods: Flynn-Wall-Ozawa (FWO) and Kissinger-Akahira-Sunose (KAS). It was concluded that the pyrolysis process of Al-lignin 20/EP and F-lignin@APP 20/EP could be divided into three stages, while the value and growth rate of the activation energy of F-lignin@APP 20/EP were much higher than that of Al-lignin 20/EP in stage III.

## 1. Introduction

Epoxy resin (EP) is widely applied in daily life, such as in insulating materials used in electronic devices, semiconductor device plastic packaging, automotive adhesives, building decoration, etc. [1,2,3,4]. There are many types of EP, among which the largest output and the most complete variety are bisphenol A epoxy resins. These account for more than 90% of the EP market share [5]. Although EP has excellent properties, such as easy curing and work, good mechanical properties, good heat resistance and solvent resistance, its poor flammability has seriously affected its use in life. The limiting oxygen index (LOI) of epoxy resin is about 20%, which means it will burn quickly after being ignited [6]. Therefore, it is very important to improve the flame retardant performance of epoxy resin.

A lot of research has been reported on the thermal behavior of epoxy resins [7,8,9], and a variety of methods have been proposed to improve its thermal performance, including the incorporation of aromatic rings and fire-retarding additives into the epoxy crosslinked network [10,11,12] or the copolymerization of epoxy resins with reactive fire retardants [13]. The thermal property and flammability of epoxy resins are strongly influenced by the chemical structure of the resins, the nature of the curing agents used and the crosslink density of the final products [9]. The aromatic epoxy resins exhibit a higher thermal stability than the aliphatic one, generally [7]. The influence of the chemical structure of an epoxy resin on its thermal stability has been reported [8]. In general, the thermal stability of a particular epoxy resin enhances with the increase of its crosslink density [9].

As the most abundant polymer in nature other than cellulose [14,15], lignin is one of the important components of the cell wall of woody plants [16,17]. The pulp produces large amounts of by-products containing lignin every year, which provides a comprehensive utilization for lignin in a large number of industries. The special chemical structure of lignin gives its unique function, and it could be added into plasticizers, solubilizers, stabilizers, flame retardants, surfactants, etc. In recent years, preparation of bio-based flame retardants with lignin has received widespread attention due to its high carbon yield after the decomposition of the aromatic framework of lignin. According to reports [18], about 35–40% of carbon would be produced after pyrolysis of lignin. Taking advantage of this feature, proper addition of lignin to polymers (such as polypropylene, PBS, PET, etc.) could effectively reduce their flammability [19]. In addition, lignin has antioxidant properties and can be added to polymer as a stabilizer to prevent the aging of the polymer. In the past two decades, there have been some reports on lignin-based flame retardants. Some of them are modified by lignin and grafted with some elements containing flame retardant effects such as P and N, while others are directly compounded with flame retardants [20,21,22,23].

Thermogravimetric (TG) analysis is a common method used to analyze biomass pyrolysis behavior and obtain kinetic parameters. Fourier transform infrared spectrometry (FTIR) can be applied to demonstrate the formation and cleavage of chemical bonds, thus indicating the completion of modification. Meanwhile, kinetics is an important factor for determining the mechanism of the thermochemical conversion of biomass. In this paper, two kinetic methods (Flynn–Wall–Ozawa (FWO) and Kissinger–Akahira–Sunose (KAS)) were applied to investigate the kinetics of a lignin-based flame retardant, and the relevant parameters were obtained [24].

## 2. Materials and Methods

### 2.1. Materials and Measurement

Alkaline lignin (Al-lignin) was purchased from TCI Shanghai Co., Ltd., Shanghai, China. Ammonium polyphosphate (APP) and 4,4-diaminodiphenylmethane (DDM) were provided by Shanghai Macklin Co., Ltd., Shanghai, China. Melamine (MEL) was provided by Shanghai Lingfeng Co., Ltd., Shanghai, China. Formaldehyde, *N*,*N*-diemthylformamide (DMF), diethyl ether and ethanol were purchased from Nanjing Chemical Reagent Co., Ltd., Nanjing, China.

TG-FTIR analysis was conducted with a PerkinElmer TGA 8000 thermogravimetric analyzer (PerkinElmer, Hopkinton, MA, USA). The X-ray photoelectron spectroscopy (XPS) spectra were recorded by an AXIS UltraDLD spectrometer with Al Kα (1486.6eV) radiation (Shimadzu/Kratos, Kanagawa, Japan). The surface morphologies of the sample were observed by using a JEOLJSM-7600 scanning electron microscopy (SEM) (JEOL, Tokyo, Japan) at the accelerating voltage of 15 kV. Limiting oxygen index (LOI) values were measured using a HC-2C oxygen index meter (Jiangsu Institute of Chemical Industry, Jiangsu, China) with a sheet size of 100 mm ×10 mm × 3 mm according to ASTM D2863-97. Vertical burning (UL-94) was conducted on a CZF-2 instrument (Jiangsu Institute of Chemical Industry, Jiangsu, China) according to ATSM D 3801 standard.

### 2.2. Sample Preparation

Firstly, Al-lignin (20 g) was firstly added to a flask containing 80 mL H_2_SO_4_ (2 mol/L) solution, heated up to 80 °C and stirred for 1.5 h. Then, when the temperature was raised to 95 °C, phenol (18 g) was added, and the mixture was stirred at reflux for 1.5 h. After completion of the reaction, the mixture was washed three times with 500 mL of diethyl ether and dried at 70 °C overnight at a constant weight.

Secondly, APP (6 g) was added to a 140 mL mixture of ethanol to water (5:2) and stirred for 30 min, being then fully dispersed in the solution. Then, MEL (5 g) was added, the temperature was raised to 70 °C and the mixture was stirred for 6 h. After completion, the mixture was cooled to room temperature, washed repeatedly with ethanol three times, and then dried at 70 °C for 12 h.

Then 4 g Ph-lignin and 20 g MELAPP were added to a flask with 250 mL DMF as solvent, then 7.2 g formaldehyde was added at 75 °C and stirred under reflux for 3 h. After completion of the reaction, 500 mL of distilled water was added, then the solid was filtered. This was washed three times with 500 mL distilled water, then dried at 70 °C under a vacuum for 24 h.

Finally, 3 g F-lignin@APP and 15 g EP were mechanically stirred for 1.5 h at 700 rpm to make the mixture uniform. After stirring, 3 g DDM was added and stirred for 1.5 h. Then, the bubbles were removed into a vacuum oven and the uniformly mixed resin was poured into the mold, cured at 100 °C for 2 h, then elevated at 150 °C for 2 h.

### 2.3. Kinetic Modeling

The kinetic analysis could be obtained through the determination of the kinetic triplets, which are the activation energy (Ea), pre-exponential factor (A) and kinetic model (f(α)). These three kinetic parameters should be determined for complete description of the kinetics for each reaction step. A large number of analytical methods are available nowadays which can be used to determine the kinetic parameters of distinct solid-phase reactions evaluation. The kinetic parameters can be determined either isothermally or nonisothermally by using two main methods: isoconversional (model-free) and model-fitting methods [25,26].

The decomposition process of lignin epoxy resin composite material is a solid state reaction process [27,28]. Generally, the rate of thermal decomposition can be expressed as a function of the degree of conversion *f*(*α*) and temperature *K*(*T*), as shown in Equation (1) [29]:(1)$$\frac{d\alpha }{dt}=\beta \frac{d\alpha }{dT}=K\left(T\right)f\left(\alpha \right)h\left(P\right)$$
where *α* is the conversion rate (%) of the sample in the reaction temperature range, *β* is the linear heating rate (°C/min), *T* is the reaction temperature (*K*) and *P* is the pressure (Pa). Although pressure may have an impact on kinetic analysis, pressure is usually ignored in pyrolysis kinetic analysis of solid state reactions [29,30]. After ignoring the pressure, Equation (1) can be written as a function containing two variables (*α* and *T*):(2)$$\frac{d\alpha }{dt}=\beta \frac{d\alpha }{dT}=K\left(T\right)f\left(\alpha \right)$$

The conversion rate (*α*) can be obtained by Equation (3):(3)$$\alpha =\left(M_{0}-M_{T}\right)/\left(M_{0}-M_{\infty }\right)$$
where *M*_0_ is the initial mass of the sample, *M**_T_* is the mass of the sample residue at the corresponding temperature and $M_{\infty }$ is the mass of the sample residue at the final temperature. The temperature-dependent equation *K*(*T*) is derived from the Arrhenius equation:(4)$$K\left(T\right)=A\exp\left(-E/RT\right)$$
where *A* and *E* are kinetic parameters. *A* is the pre-exponential factor, *E* is apparent activation energy (kJ/mol) and *R* is the gas constant (8.314 J/(mol·K)).

Bring Equation (4) into Equation (1), separate the variables and integrate the equation to get Equation (5):(5)$$G\left(\alpha \right)=\int_{0}^{\alpha }d\alpha /f\left(x\right)=\frac{A}{B}\int_{0}^{T}\exp\left(-E/RT\right)dT=\left(AE/\beta R\right)P\left(u\right)$$
where *G*(*α*) is the integral form of *f*(*x*), *u* is usually defined as the temperature integral of *E*/*RT* and *P*(*u*). *P*(*u*) has many mathematical assumptions to determine the activation energy.

In the absence of a kinetic model, there are several ways to obtain the activation energy during the reaction. Usually these methods are divided into the differential method and integral method. The integral method usually deviates greatly from the actual value. Compared with the integral method, the differential method does not involve approximate values, so the results obtained are more accurate.

Kissinger–Akahira–Sunoe (KAS) and Flynn–Wall–Ozawa (FWO) are two methods commonly used to obtain the activation energy of thermal decomposition:(6)$$\ln\left(\beta /T^{2}\right)=\ln\left(AR/EG\left(\alpha \right)\right)-E/RT$$
(7)$$\ln\left(\beta \right)=\ln\left(AR/EG\left(\alpha \right)\right)-2.315-0.4567E/RT$$

According to the KAS and FWO methods, the activation energy for a specific value can be obtained from the slopes of plot ln(*β*/*T*^2^) vs. 1/*T* and plot log (*β*) vs. 1/*T*, respectively, for a set of 4 heating rates (*β*). It should be noted that *T* is strongly dependent on *β* at a specific value, as discussed previously.

### 2.4. Characterization

Fourier transform infrared (FTIR) spectra were recorded by a Nicolet 6700 FTIR spectrometer under the resolution of 1 cm⁻^1^ in 32 scans by a KBr disk with the wavenumber ranging from 4000 to 500 cm^−1^. Thermogravimetric analysis (TG) was obtained on a DTG-60AH (SHIMADZU, Japan). The epoxy resin composites (about 5 mg) were heated from 30 to 700 °C at a rate of 15 °C min^−1^ under the nitrogen flow of 40 mL·min⁻^1^. The TG–FTIR analysis was conducted on a thermogravimetric analysis (TGA, Perkin Elmer Pyris 1 TGA unit), coupled with a Fourier transform infrared spectrometer. The X-ray photoelectron spectroscopy (XPS) spectra were recorded by an AXIS UltraDLD spectrometer with Al Kα (1486.6 eV) radiation.

## 3. Results and Discussion

### 3.1. Characterization of F-Lignin@APP

The FTIR spectra of Al-lignin, Ph-lignin and F-lignin@APP are shown in Figure 1. From Figure 1, we can observe that the basic structure of lignin does not change much after phenolation. Aromatic skeletal stretching vibrations shift from 1600 and 1503 cm^−1^ to 1624 and 1512 cm^−1^, C–H bending and out-of-plane deform vibrations shift from 1460 and 858 cm^−1^ to 1458 and 873 cm^−1^ [31]. The peak at 1222 cm^−1^ is assigned to the syringyl structure. Its absorption peak intensity is enhanced, indicating that the phenolization of lignin was successful [32]. After F-lignin@APP wrapped APP, some absorption peaks appeared, such as the absorption peak of the N–H bending vibration in the NH_2_ structure at 1640 cm⁻^1^, the absorption peak of the triazine ring at 813 cm⁻^1^ and the absorption peak of the stretching vibration P–O at 1088 cm⁻^1^ [33].

The chemical composition and types of chemical bonds can be analyzed by XPS spectra. The N_1s_ spectra of the MEL and F-lignin@APP are shown in Figure 2. It can be seen from Figure 2a that the peak at 401.6 eV represents NH_2_ in the MEL and the peak at 399.3 eV may stand for the nitrogen in the C=N double bonds of the MEL [34]. As can be observed from Figure 2b, the peak of C=N double bond in the MEL still appears at 399.3 eV. After functionalization, two new peaks appeared; the peak at 401.2 eV corresponds to NH_4_^+^ in the APP. Another peak at 398.1 eV should be attributed to the N in the -NH– for F-lignin@APP [35].

The thermogravimetry (TG) and derivative thermogravimetry (DTG) curves of Al-lignin, Ph-lignin and F-lignin@APP are shown in Figure 3. As can be observed from Figure 3a, the carbon residues of the Ph-lignin and F-lignin@APP were reduced compared with the Al-lignin: i.e., 33.5% and 37.5%, respectively. In addition, the DTG curves of the three samples all showed a peak before 150 °C due to evaporation of the residual moisture contained in the three samples. Different from the Al-lignin and F-lignin@APP, a broad endothermic peak was observed for the Ph-lignin at 160–320 °C, which might be formed by the decomposition and volatilization of lignin with lower molecular weight. At 320–500 °C, an endothermic peak appeared for the three samples, because the bonds between the structural units of lignin were broken and some monomer phenols evaporated. In addition, it could be found that the endothermic peak of the F-lignin@APP is much stronger than the other samples due to the degradation of the lignin and the decomposition of the APP during the heating process.

The FTIR spectra of volatile products originating from Al-lignin, Ph-lignin and F-lignin@APP are shown in Figure 4. The characteristic peaks of F-lignin@APP are indicated at 3330, 2356, 1593, 1509, 1100, 962 and 928 cm⁻^1^. Among them, the peaks at 3330 cm^−1^ and 1593 cm^−1^ are absorption peaks of NH_3_. The peaks at 1100, 962 and 928 cm^−1^ illustrate the presence of phosphoric acid derivatives, which are attributed to P–O–C, P–O–Ph and P–O–P, respectively [20,36]. Compared with Ph-lignin and Al-lignin, the absorption peaks of F-lignin@APP do not contain hydrocarbon compounds, carbonyl compounds and aromatic compounds, which are usually formed by breaking the chemical bonds of functional groups on the basic structural units of lignin. It was concluded that the depolymerization products of lignin synergized with the polyacid generated by APP decomposition would form phosphorus-containing derivatives.

### 3.2. Thermal and Fire Properties of F-Lignin@APP 20/EP

The degradation process of EP, Al-lignin 20/EP and F-lignin@APP 20/EP under an N_2_ atmosphere is shown in Figure 5 and the detailed data are listed in Table 1. From Figure 5, the T_dmax_ (the temperature of the maximum decomposition rate) of EP, Al-lignin 20/EP and F-lignin@APP 20/EP are 405, 401 and 382 °C, respectively. Due to the presence of APP in the flame retardant, the degradation temperature of APP is lower than that of F-lignin@APP. The maximum decomposition rate of F-lignin@APP 20/EP is 11.8%/min, which is 8%/min and 4.7%/min lower than EP and Al-lignin. In addition, the char residue of F-lignin@APP 20/EP is 32.5%, which is much higher than EP. Lower decomposition rate and higher char residue might effectively improve the thermal stability of EP modified by F-lignin@APP.

In order to investigate the flame retardance of the EPs, the LOI tests and vertical burning tests (UL-94) were carried out at the room temperature, and the results are shown in Table 2. As shown in Table 2, the LOI of EP is only 26.7%. As the content of F-lignin@APP increases, the LOI of the EP reaches 34.7% and a V1 classification (good flame retardancy) at the thickness of 3 mm. While the content of F-lignin@APP reaches 20 wt %, the result of the LOI test indicates that the F-lignin@APP20/EP gets an LOI (36.1%) that is much higher than the LOI of EP. Moreover, it is easy to reach a V0 classification (excellent flame retardancy) with 20 wt % of the F-lignin@APP added.

### 3.3. Characterization of Gas Phase

FTIR spectra of the volatilities of EP, Al-lignin 20/EP and F-lignin@APP 20/EP at T_dmax_ are shown in Figure 6. We can observe that they have several of the same absorption peaks: e.g., the peak near 3650 cm^−1^ is the absorption peak of NH_3_ and H_2_O alike. The peak at 2975 cm^−1^ is attributed to the stretching vibration absorption peak of the hydrocarbon [31]. The peak at 2358 cm^−1^ is assigned to the stretching vibration absorption peak of the C=O in the CO_2_. The peak at 1750 cm^−1^ is the absorption peak of the carbonyl component, and the peaks near 1515 cm^−1^ are the absorption peaks of the aromatic component [37]. The peaks around 1170 cm^−1^ are mainly the absorption peaks of the C–C and C–O originating from the structure of the phenols and ethers. These absorption peaks are closely related to the epoxy resin, i.e., bisphenol A type, and the use of the amino group containing a generated curing agent (DDM) [38]. Unlike Al-lignin 20/EP and EP, F-lignin@APP 20/EP has several new absorption peaks. The peak at 3330 cm^−1^ is the absorption peak of NH_2_ in the melamine. The peaks at 962 and 928 cm^−1^ are the absorption peak of the phosphorus-containing derivatives, P–O–P and P–O, respectively. In addition, the absorption peak intensity of the carbonyl compound decreases. It is indicated that F-lignin@APP 20/EP can produce polyacids and NH_3_ during pyrolysis. These polyacids interact with the carbon produced by the decomposition of lignin to form a condensed phase, which could block some of the volatiles to some extent.

### 3.4. Kinetics Analysis

Al-lignin 20/EP (Figure 7) and F-lignin@APP20/EP (Figure 8) pyrolysis kinetic parameters were estimated by KAS method and FWO method. The activation energy at different conversion rates obtained from both methods is shown in Table 3and Table 4. The coefficient (R^2^) of the composite materials methods was around 0.99, which meant the data fit well. In addition, the difference in activation energy data obtained by KAS and FWO methods was very small and this means that the estimated value was relatively reliable.

The activation energy of Al-lignin 20/EP and F-lignin@APP 20/EP at different conversion rates is shown in Figure 9. It can be seen that the thermal decomposition of both materials is a very complicated process, accompanied by many reactions. Combined with the gas phase analysis, the pyrolysis process of Al-lignin 20/EP and F-lignin@APP 20/EP can be divided into three stages. In stage I, the activation energy required for pyrolysis is low. During this stage, the activation energy required for the cleavage of some functional groups in the lignin structure, e.g., hydroxyl, methyl and methoxy groups, is low [39]. In addition, the APP might decompose, producing some phosphorus-containing derivatives and NH_3_ at this stage. In stage II, the lignin will depolymerize at a higher temperature. The thermal stability of the ether bond is lower than that of the C–C bond, so this kind of bond will be broken first [40], producing more ether compounds in volatiles. In addition, the decomposition products of the APP will also interact with the depolymerization products of the lignin at this stage. Therefore, the growth rate of the activation energy of F-lignin@APP 20/EP is higher than that of Al-lignin 20/EP, as shown in Figure 9. In stage III, the growth rate of the activation energy of F-lignin@APP 20/EP is much higher than that of Al-lignin 20/EP, indicating that the carbon layer formed by F-lignin@APP 20/EP is more stable than Al-lignin 20/EP.

## 4. Conclusions

A novel lignin-based flame retardant (F-lignin@APP) containing P and N was successfully prepared. TG–DTG results showed that the maximum decomposition rate of F-lignin@APP 20/EP was 11.8%/min, which was 8%/min and 4.7%/min lower than that of EP and Al-lignin. In addition, the char residue of F-lignin@APP 20/EP was 32.5%, much higher than that of EP. Moreover, the pyrolysis process of Al-lignin 20/EP and F-lignin@APP 20/EP can be divided into three stages. F-lignin@APP played an important part during stage III, leading to a good flame retardant performance due to the carbon layer formed at a high temperature.

## Figures and Tables

**Figure 1 polymers-12-02123-f001:**
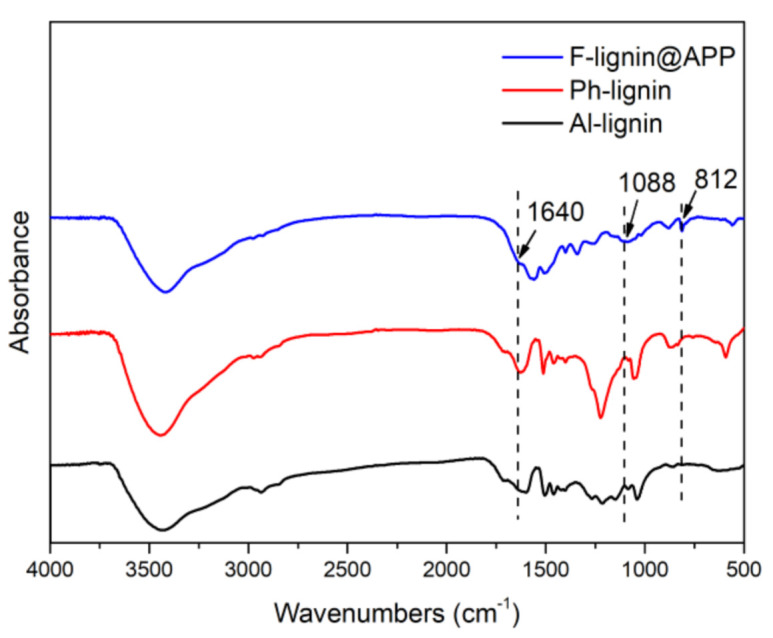
FTIR spectra of Al-lignin, Ph-lignin and F-lignin@APP.

**Figure 2 polymers-12-02123-f002:**
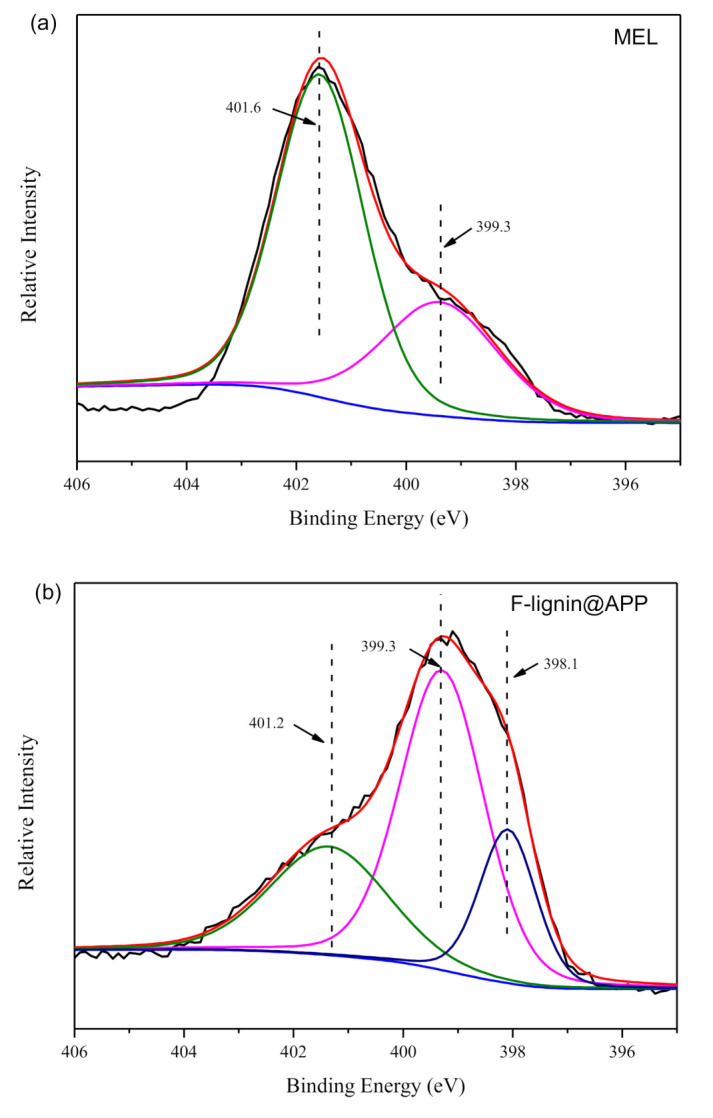
N_1s_ spectra of (**a**) melamine (MEL) and (**b**) F-lignin@APP.

**Figure 3 polymers-12-02123-f003:**
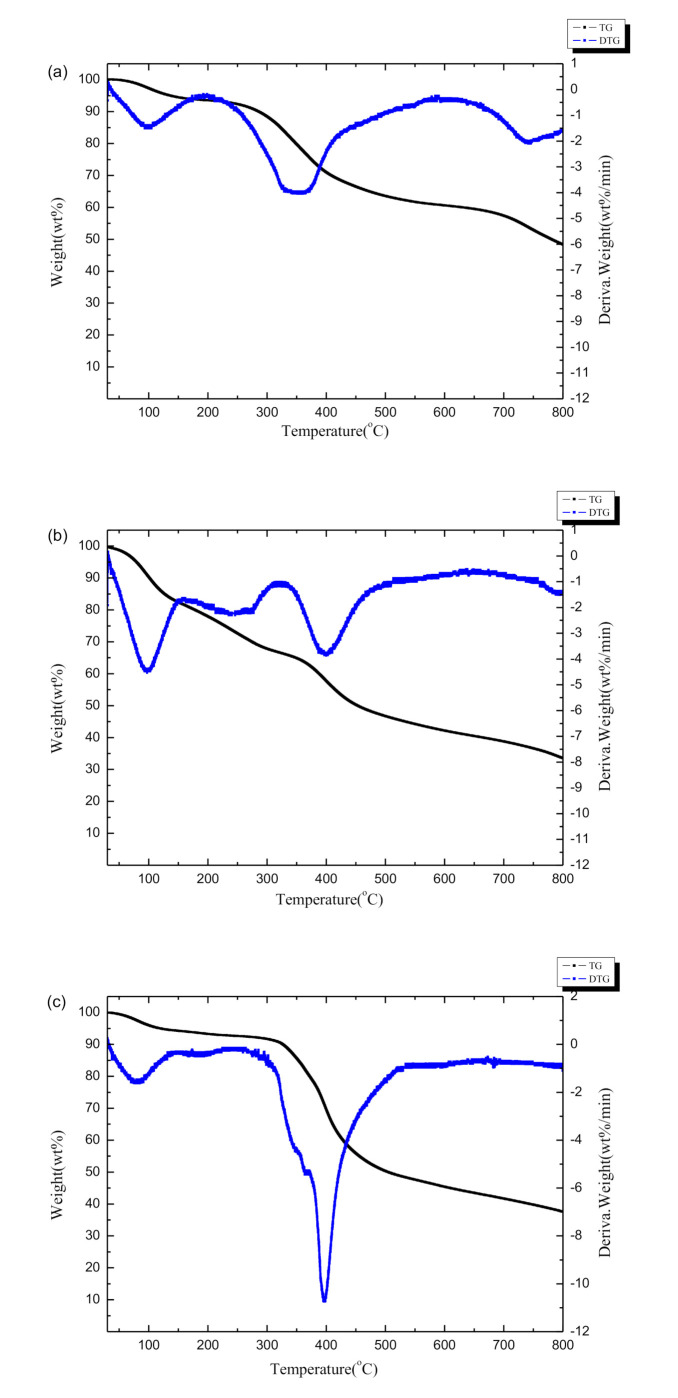
Thermogravimetry (TG) and derivative thermogravimetry (DTG) curves of (**a**) Al-lignin, (**b**) Ph-lignin and (**c**) F-lignin@APP.

**Figure 4 polymers-12-02123-f004:**
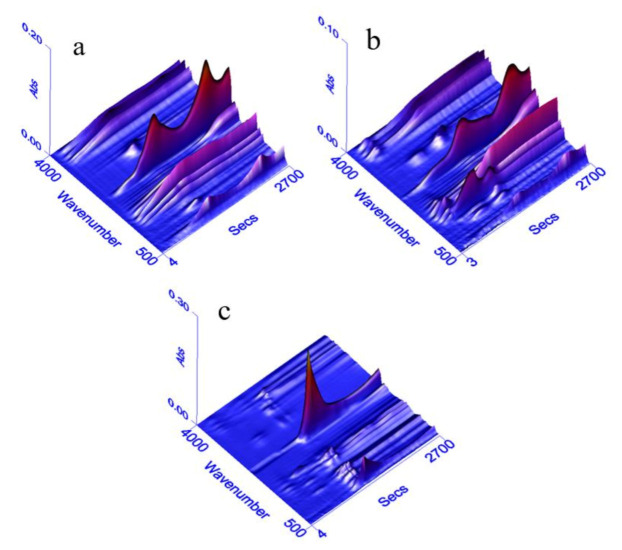
3D FTIR spectra of the volatile products of Al-lignin (**a**), Ph-lignin (**b**) and F-lignin@APP (**c**).

**Figure 5 polymers-12-02123-f005:**
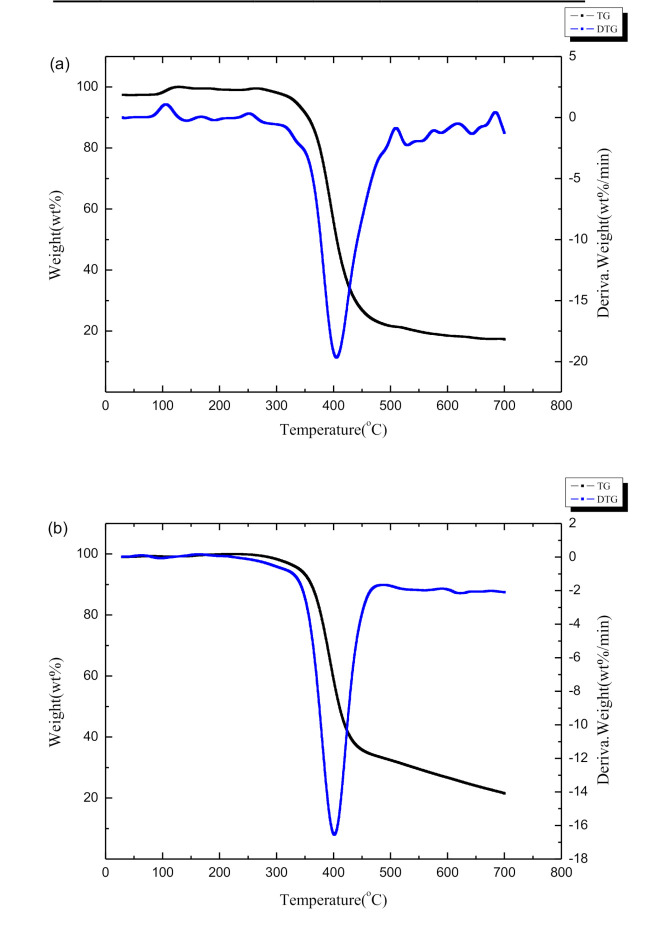
TG and DTG curves of (**a**) EP, (**b**) Al-lignin20/EP and (**c**) F-lignin@APP20/EP.

**Figure 6 polymers-12-02123-f006:**
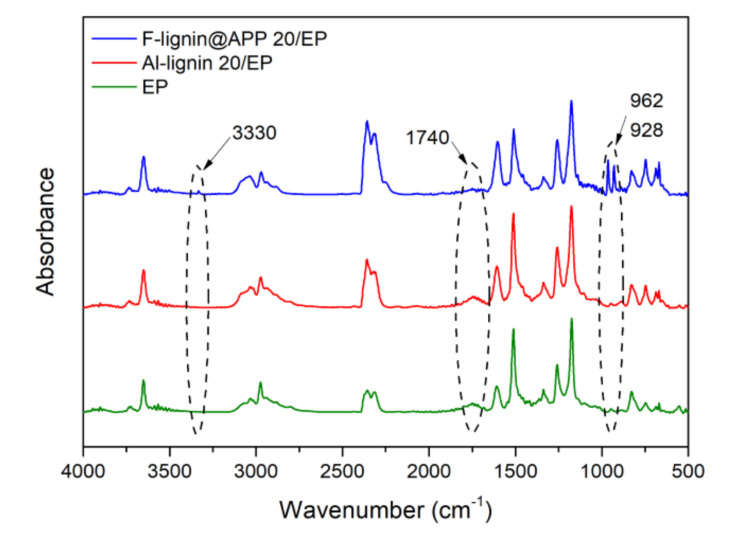
FTIR spectra of the volatile products of EP, Al-lignin 20/EP and F-lignin@APP20/EP.

**Figure 7 polymers-12-02123-f007:**
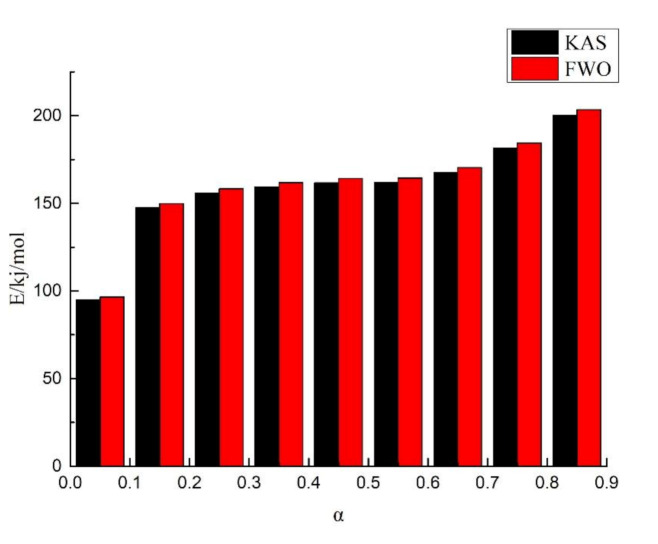
Histogram of Kissinger–Akahira–Sunose (KAS) method and Flynn–Wall–Ozawa (FWO) method for Al-lignin 20/EP.

**Figure 8 polymers-12-02123-f008:**
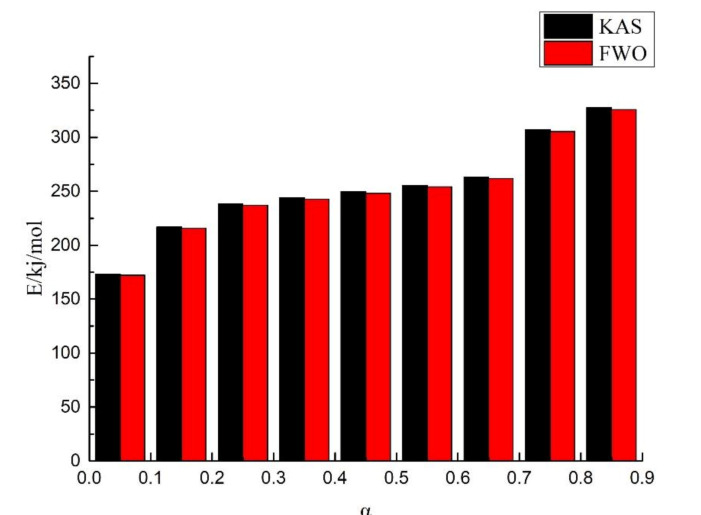
Histogram of KAS method and FWO method for F-lignin@APP20/EP.

**Figure 9 polymers-12-02123-f009:**
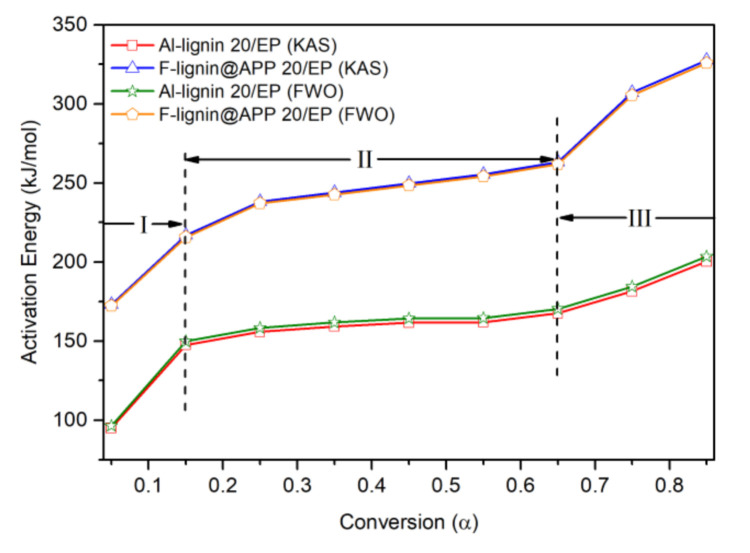
Activation energy of Al-lignin 20/EP and F-lignin@APP20/EP at different conversion.

**Table 1 polymers-12-02123-t001:** Thermogravimetric analysis (TGA) data of epoxy resin (EP), Al-lignin 20/EP and F-lignin@APP20/EP in N_2_.

Samples	T_d5%_	T_dmax_	dW/dT	Residues
	(°C)	(°C)	(%/min)	(%)
EP	333	405	19.8	17.2
Al-lignin 20/EP	340	401	16.5	21.5
F-lignin@APP20/EP	325	382	11.8	32.5

**Table 2 polymers-12-02123-t002:** Flame retardance of EP, F-lignin@APP18/EP and F-lignin@APP20/EP.

Sample	EP	F-Lignin@APP18/EP	F-Lignin@APP20/EP
LOI (%)	26.7	34.7	36.1
UL-94 rating	-	V1	V0

**Table 3 polymers-12-02123-t003:** Activation energy of Al-lignin 20/EP obtained from the KAS method and FWO method.

α	KAS (R^2^)	FWO (R^2^)
0.05	95.00 (0.98)	96.52 (0.99)
0.15	147.49 (0.98)	149.87 (0.99)
0.25	155.76 (0.99)	158.26 (0.99)
0.35	159.29 (0.99)	161.84 (0.99)
0.45	161.66 (0.99)	164.25 (0.99)
0.55	161.94 (0.99)	164.54 (0.99)
0.65	167.66 (0.99)	170.36 (0.99)
0.75	181.48 (0.99)	184.40 (0.99)
0.85	200.22 (0.98)	203.42 (0.99)

**Table 4 polymers-12-02123-t004:** Activation energy of F-lignin@APP20/EP obtained from the KAS method and FWO method.

α	KAS (R^2^)	FWO (R^2^)
0.05	173.28 (0.99)	172.39 (0.99)
0.15	217.00 (0.97)	215.80 (0.98)
0.25	238.31 (0.99)	237.11 (0.99)
0.35	243.94 (0.99)	242.71 (0.99)
0.45	249.63 (0.99)	248.37 (0.99)
0.55	255.39 (0.99)	254.11 (0.99)
0.65	263.17 (0.99)	261.85 (0.99)
0.75	307.06 (0.98)	305.36 (0.98)
0.85	327.45 (0.99)	325.80 (0.99)

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
