# Peer review of "Thermal Kinetics of a Lignin-Based Flame Retardant"

_polymers, 2020, doi:10.3390/polym12092123_

Round 1
Reviewer 1 Report
Recommendation: Publish after minor revisions noted.
Liang et al. described Thermal kinetics of a lignin-based flame retardant. This paper is appropriate for Polymers, though the following minor comments should be addressed.
It,s mentioned in the abstract that to improve the flame retardant performance of epoxy resin (EP), a lignin-based flame retardant was prepared to achieve it. However, in the manuscript, they did provide any data to assess any improvement regarding flame retardant performance example, LOI, burning test.
In the methods, the authors mention in Materials and Methods section 2.2. Phenolation of alkaline lignin (Ph-lignin), 2.3. Preparation of MEL and APP mixture (MELAPP), 2.4. Preparation of lignin-based flame retardant (F-lignin@APP); however, no characterization provided (NMR, FTIR).
Author Response
Dear Editors:
We have revised the manuscript according to reviewers' comments. In our point-by-point response attached below, reviewer comment is in Black and our response is in Blue.
We look forward to hearing from you. Thank you.
Note: In order to highlight the changes what we have done, the color of text changed will become yellow.
Following are our response about reviewers' comments to our manuscript:
------------------------------------------------------
Reviewer 1
Comments:
- It’s mentioned in the abstract that to improve the flame retardant performance of epoxy resin (EP), a lignin-based flame retardant was prepared to achieve it. However, in the manuscript, they did provide any data to assess any improvement regarding flame retardant performance example, LOI, burning test.
Answer: Revised. We had data about the flame retardant of the EP and F-lignin@APP/EP, but this manuscript was mainly focused on thermal kinetics of F-lignin@APP/EP, so we didn’t mention any data such as LOI and burning test. Now we have listed data of LOI and burning test (see Table 2).
- In the methods, the authors mention in Materials and Methods section 2.2. Phenolation of alkaline lignin (Ph-lignin), 2.3. Preparation of MEL and APP mixture (MELAPP), 2.4. Preparation of lignin-based flame retardant (F-lignin@APP); however, no characterization provided (NMR, FTIR).
Answer: Revised. We added XPS data to analyze the MEL and F-lignin@APP in Figure 2.
------------------------------------------------------
Xiaoli GU
Nanjing Forestry University
Reviewer 2 Report
I reviewed the article entitled “Thermal kinetic of a lignin-based flame retardant” by Liang et al.
I am little skeptical about this article, the authors did some effort in chemical analysis of the material but the work is presented somehow disjoint with significant language problem. I suggest they should consult someone who feels comfortable to write technical paper in English, or maybe getting some professional help from MDPI editorial division.
There are lots of redundancies in sentences, for example , line 11 flame retardant is used twice in one sentence.
Line 13 This paper was to investigate ???
Line 15 was enhanced and it means
Line 14 FT-IR result (should be results) results of what ???
Introduction is not acceptable , it is very short and superficial , there is no justification of the work what so ever.
2.1 through 2.6 can be combined seamlessly .
again language problem, line 62 Melamine (MEL) were offered ?????
There are many more language problems which makes awkward to read the manuscript. As you can imagine I can give all here , I tried to pin point only some as example.
Line 89 -117 kinetic modeling is standing alone.
2.7 Measurement of what ???
Result are presented but I don’t see much clear discussion.
Line 202 (see Figure 6) don't use such format , as illustrated in Figure 6
Line 203 ……rates……..from both methods was shown ?????
Line 205 ….which showed that the estimated value was relatively reliable ??????
Conclusions
You never start conclusion as “In summary”
In retrospect, in my opinion I am afraid article is not acceptable as it is unless a significant revision is carried out before it is considered for publication in Polymers.
Author Response
Dear Editors:
We have revised the manuscript according to reviewers' comments. In our point-by-point response attached below, reviewer comment is in Black and our response is in Blue.
We look forward to hearing from you. Thank you.
Note: In order to highlight the changes what we have done, the color of text changed will become yellow.
Following are our response about reviewers' comments to our manuscript:
------------------------------------------------------
Reviewer 2
Comments:
- There are lots of redundancies in sentences, for example , line 11 flame retardant is used twice in one sentence.
Answer: Revised. The first “flame retardant” was revised to “thermal property”.
- Line 13 This paper was to investigate ???
Answer: Revised.
- Line 15 was enhanced and it means
Answer: Revised.
- Line 14 FT-IR result (should be results) results of what ???
Answer: Revised. We added “the peak at 1222cm-1 and it was assigned to syringyl structure”.
- Introduction is not acceptable , it is very short and superficial , there is no justification of the work what so ever.
Answer: Revised. We added a segment about the EP thermal behavior and flammability.
- 2.1 through 2.6 can be combined seamlessly .
Answer: Revised. The preparations were combined.
- again language problem, line 62 Melamine (MEL) were offered?????
Answer: Revised. “MEL were offered” was changed to “MEL was offered”.
- There are many more language problems which makes awkward to read the manuscript. As you can imagine I can give all here , I tried to pin point only some as example.
Answer: Thanks for your advice. And we have updated the whole manuscript.
- Line 89 -117 kinetic modeling is standing alone.
Answer: Revised.
- 2.7 Measurement of what ??? Result are presented but I don’t see much clear discussion.
Answer: Revised.
- Line 202 (see Figure 6) don't use such format , as illustrated in Figure 6
Answer: Revised. (see Figure 6) was changed to (Figure 7), (see Figure 7) was changed to (Figure 8).
- Line 203 ……rates……..from both methods was shown ?????
Answer: Revised.
- Line 205 ….which showed that the estimated value was relatively reliable ??????
Answer: Revised. We revised: the difference in activation energy data obtained by KAS and FWO methods was very small and it mean that the estimated value was relatively reliable.
- Conclusions You never start conclusion as “In summary”
Answer: Revised.
------------------------------------------------------
Xiaoli GU
Nanjing Forestry University
Round 2
Reviewer 2 Report
Some enhancement of the manuscript has been carried out by the authors I guess it can be published as it is.